# Current Perspectives on the Auxin-Mediated Genetic Network that Controls the Induction of Somatic Embryogenesis in Plants

**DOI:** 10.3390/ijms21041333

**Published:** 2020-02-16

**Authors:** Anna M. Wójcik, Barbara Wójcikowska, Małgorzata D. Gaj

**Affiliations:** Institute of Biology, Biotechnology and Environmental Protection, Faculty of Natural Sciences, University of Silesia in Katowice, Jagiellońska 28, 40-032 Katowice, Poland; malgorzata.gaj@us.edu.pl

**Keywords:** auxin, Aux/IAA, auxin signalling, ARFs, LAFL group genes, somatic embryogenesis, transcription factors

## Abstract

Auxin contributes to almost every aspect of plant development and metabolism as well as the transport and signalling of auxin-shaped plant growth and morphogenesis in response to endo- and exogenous signals including stress conditions. Consistently with the common belief that auxin is a central trigger of developmental changes in plants, the auxin treatment of explants was reported to be an indispensable inducer of somatic embryogenesis (SE) in a large number of plant species. Treating in vitro-cultured tissue with auxins (primarily 2,4-dichlorophenoxyacetic acid, which is a synthetic auxin-like plant growth regulator) results in the extensive reprogramming of the somatic cell transcriptome, which involves the modulation of numerous SE-associated transcription factor genes (*TF*s). A number of SE-modulated TFs that control auxin metabolism and signalling have been identified, and conversely, the regulators of the auxin-signalling pathway seem to control the SE-involved TFs. In turn, the different expression of the genes encoding the core components of the auxin-signalling pathway, the AUXIN/INDOLE-3-ACETIC ACIDs (Aux/IAAs) and AUXIN RESPONSE FACTORs (ARFs), was demonstrated to accompany SE induction. Thus, the extensive crosstalk between the hormones, in particular, auxin and the TFs, was revealed to play a central role in the SE-regulatory network. Accordingly, LEAFY COTYLEDON (LEC1 and LEC2), BABY BOOM (BBM), AGAMOUS-LIKE15 (AGL15) and WUSCHEL (WUS) were found to constitute the central part of the complex regulatory network that directs the somatic plant cell towards embryogenic development in response to auxin. The revealing picture shows a high degree of complexity of the regulatory relationships between the TFs of the SE-regulatory network, which involve direct and indirect interactions and regulatory feedback loops. This review examines the recent advances in studies on the auxin-controlled genetic network, which is involved in the mechanism of SE induction and focuses on the complex regulatory relationships between the down- and up-stream targets of the SE-regulatory TFs. In particular, the outcomes from investigations on Arabidopsis, which became a model plant in research on genetic control of SE, are presented.

## 1. Introduction

The process of somatic embryogenesis (SE) demonstrates the unique developmental capacity of plants for switching on the embryogenic programme of development in somatic cells that have already differentiated. As a result of the embryogenic transition, bipolar structures that resemble zygotic embryos, which are called somatic embryos, are formed. Somatic embryos are able to develop into complete plants and since 1960s, numerous protocols have been established that enable the efficient regeneration of dozens of plant species via somatic embryos that are induced under in vitro culture conditions (reviewed in [1]). In addition to its wide implementation in the mass micropropagation of plants for the market, the SE-based protocols are also commonly used to produce transgenic plants (reviewed in [2]). In addition to its practical significance in plant biotechnology, the SE process provides a valuable model system for studies on plant embryogenesis and the developmental plasticity of the plant somatic cells [3,4]. Identifying the exo- and endogenous factors that promote embryogenic transition in in vitro-cultured somatic cells leads to a better understanding of plant cell totipotency. In contrast to advanced knowledge on the tissue culture conditions that promote SE induction [5], the endogenous factors that determine the tissue capacity for an embryogenic response are much less recognised. However, distinct progress in deciphering the molecular mechanism that governs the embryogenic transition of plant somatic cells has been made in the last decade and the role of auxin in SE induction has been intensively studied at the molecular level.

## 2. Auxin, a Main Inducer of SE

Since the classical demonstration of SE induction in carrot callus [6], the impact of auxin on the embryogenic pathways that are induced in in vitro-cultured plant tissues has become evident and hundreds of protocols have been established to induce SE in a plethora of plant species [7]. Although there are some differences between the in vitro culture conditions that are applied to induce SE in different plants, treating explants with auxin and auxin-like substances seems to be commonly required for SE induction. In support of this statement, auxin treatment was applied in 74 of the 80 protocols reviewed in Appendix A and auxin was frequently combined with cytokinins (65%). The most effective and commonly used inducer of SE (78% of the protocols) is 2,4-dichlorophenoxyacetic acid (2,4-D), which is a synthetic plant growth regulator of auxin activity.

## 3. Auxin- and Stressor-Like Activity of 2,4-D in SE Induction

2,4-D was initially used in cereal agriculture as a systemic herbicide that selectively kills broadleaf weeds [8]. Treatment with 2,4-D causes a number of morphological changes in plants in a concentration-dependent manner. 2,4-D efficiently stimulates cell division, plant growth and in vitro-induced morphogenetic responses at low concentrations of 1–10 mg/L, while spraying the plants in the field with 2,4-D of a concentration of about one-thousand-fold higher (2 g/L, according to the weed control manuals) caused lethal symptoms that were similar to an auxin overdose [9]. Of note, in spite of the distinctly higher tolerance of the monocot vs. the dicot plants to the toxic effects of 2,4-D [10], similar concentrations of this substance are applied to induce SE in different plants, and relevantly, 1–2 mg/L of 2,4-D is recommended for SE induction in maize and wheat [11,12,13,14], while 1.1 mg/L of 2,4-D is effective for establishing an embryogenic culture in Arabidopsis [15].

2,4-D shares structural similarity with indole-3-acetic acid (IAA), which involves the presence of a carboxyl group and an aryl ring structure [10]. However, in contrast to natural auxin, IAA, which is rapidly inactivated via conjugation and degradation in plant cells [16], the effects of 2,4-D on plants are long-lasting because of its high level of stability in the plant cells [17]. Moreover, the polar cell-to-cell transport of 2,4-D and IAA differs in terms of the auxin carriers that are engaged. Accordingly, in contrast to IAA, which is transported polarly by both types of auxin carriers, i.e., the efflux PIN-FORMED (PIN) and *P-*GLYCOPROTEIN (PGP) and the influx AUXIN1/LIKE-AUX1 (AUX/LAX) proteins, the transport of 2,4-D by the PIN efflux carriers is limited, which results in an accumulation of 2,4-D in plant cells [18]. Moreover, 2,4-D has a weak binding affinity to the IAA receptor, AUXIN-BINDING PROTEIN1 (ABP1) [19]. However, similar to IAA, 2,4-D is able to promote the binding of AUXIN/INDOLE-3-ACETIC ACID (Aux/IAA) proteins to the TRANSPORT INHIBITOR RESPONSE1 (TIR1) F-box protein of the auxin-signalling pathway after the auxin responsive genes are activated by the AuxRE (AUXIN RESPONSIVE ELEMENT) motif in the promotors [20,21]. It is worth noting that the 2,4-D and IAA-induced transcriptomes tend to overlap only to some extent [20], which reflects the differences between IAA and synthetic auxin in their interactions with the signalling machinery [22].

In addition to its auxin-like activity, 2,4-D is believed to affect the plant developmental processes by inducing the stress responses. Accordingly, treatment with 2,4-D causes an accumulation of ROS (reactive oxygen species) and the stress hormones ethylene and abscisic acid (ABA) in plant tissue [23,24,25]. Relevant to the stress-related mechanism of 2,4-D activity, the central role of stress factors, in particular, oxidative stress, in promoting the developmental switches in plant cells including the induction of SE has been widely demonstrated (reviewed in [26,27]). In support of the stressor-like activity of 2,4-D, numerous stress-responsive genes were found to be differently expressed in 2,4-D-induced SE-transcriptomes [28,29,30,31,32,33].

However, a long-lasting debate on the auxin- vs. stress-like activity of 2,4-D seems to be groundless given that the complex relationships between auxin and stress responses in controlling plant development have been documented, and relevantly, different stress factors impact auxin signalling and biosynthesis [34] (reviewed in [35]). Moreover, 2,4-D treatment also activated the core regulators of the auxin-signalling AUXIN RESPONSE FACTORs (ARFs) and the TRYPTOPHAN AMINOTRANSFERASE OF ARABIDOPSIS1/YUCCA, TAA1/YUC-dependent auxin biosynthesis pathway in SE-induced explants of Arabidopsis [36,37]. Thus, both the auxin- and stressor-like intrinsic function of 2,4-D are closely related in promoting embryogenic development in plant cells.

## 4. Biosynthesis and Accumulation of Auxin During SE

The observation that the embryogenic explants of various plants contain a higher level of IAA than the non-embryogenic explants implies that there is a positive relationship between the auxin content and the capacity of tissue to induce SE [38,39,40,41,42,43,44,45]. However, in conifers, the SE-recalcitrant cultures had a higher level of IAA [46,47], which suggests that the auxin content that enables SE induction seems to be genotype and tissue specific. It seems that not the initial level of IAA in explant tissue but rather a dynamic increase of the endogenous auxin content in response to the SE-induction factors might provide an indispensable signal for the embryogenic transition. Relevantly, a significant increase in the IAA content was reported to accompany early SE induction in different plants including carrot [48], sunflower [49], robusta coffee [50,51,52], alfalfa [53], pineapple guava [54], Arabidopsis [36], cotton [55] and Norway spruce [56]. Besides in vitro culture conditions, transgenic plants of Arabidopsis that produced somatic embryos in planta as a result of the overexpression of the *LEAFY COTYLEDON2* (*LEC2*) gene were also found to accumulate IAA [57]. Auxin accumulation in plant tissue in response to the SE-induction signal resembles the early stages of ZE. Accordingly, an increase in the endogenous auxin level was observed in the early-stage zygotic embryos of carrot [58] and other plants [54]. In the ZE of Arabidopsis, IAA is maternally biosynthesised de novo in the fertilised ovules and provides a source of auxin for the correct embryo development [59,60]. The auxin accumulation that accompanies the embryogenic development that is induced in both generative and somatic plant cells provides further support for a similarity in the molecular and hormonal signals that control embryogenic development in vivo and in vitro [3,61].

Two alternative metabolic pathways, the tryptophan-dependent and -independent control de novo IAA biosynthesis in plant development including zygotic embryo development [59,62,63]. In an SE culture of Arabidopsis, a TAA1/YUC pathway, which is controlled by the tryptophan aminotransferases TAA1 and TRYPTOPHAN AMINOTRANSFERASE RELATED1-4 (TAR) and YUCCA flavin-dependent monooxygenases, has been identified [36]. Six of eleven *YUC* genes, *YUC1, YUC2, YUC4, YUC6, YUC10* and *YUC11*, were found to be expressed in different embryogenic cultures of Arabidopsis [36,64,65]. It is worth noting that the SE-associated *YUC*s were specifically expressed in the shoot part of a plant and that most of them were active in ZE [66,67]. The expression of the SE-involved *YUC*s was localised in the SE-involved explant part, i.e., in the cotyledons of the immature zygotic embryo, and that the relevant single and multiple *yuc* mutants, including *yuc2, yuc4* and *yuc1, yuc4, yuc10* and *yuc11* had a severely reduced capacity for SE [36,65,68]. At the transcriptional level, auxin biosynthesis is regulated by the developmental and environmental signals that control the binding of specific TFs to the auxin biosynthesis-related genes including the *YUC*s [69]. Thus, the set of SE-involved *YUC* genes might differ among the plants, explants and treatment that is applied. Accordingly, in Arabidopsis, the 2,4-D treatment of immature zygotic embryos (IZE) explants activated *YUC1, YUC4* and *YUC10* [36], while only two of these genes were stimulated in the developing somatic embryos of the trichostatin A-induced IZE explants [68]. Ethylene was indicated to control the *YUC* expression and the negative impact of ethylene on SE induction in Arabidopsis was attributed to the inhibition of the *YUC*s, and subsequently, a disturbed auxin accumulation in the culture [65].

In line with the results on Arabidopsis, a RNA-seq analysis of the SE-transcriptome in cotton revealed the up-regulation of numerous genes that are involved in the tryptophan-dependent auxin biosynthesis, including the *YUCs*, *TRP1* (*TRYPTOPHAN BIOSYNTHESIS1*), *ASB1* (*ANTHRANILATE SYNTHASE*), *TSB2* (*TRYPTOPHAN SYNTHASE Β-SUBUNIT2*), *NIT4* (*NITRILASE4*) and *CM1* (*CHORISMATEMUTASE*) [70,71]. An increase of the *YUC1* and *TAA1* transcripts that correlated with an increased level of endogenous IAA was also observed in the embryogenic explants of coffee [50]. Thus, the tryptophan-dependent TAA1-YUC pathway of IAA biosynthesis seems to commonly contribute to the auxin accumulation that is associated with the embryogenic transition in the somatic cells of plants. However, insights into the available RNA-seq data of an *O. sativa* embryogenic culture [72] indicated only a limited number of auxin biosynthesis-related transcripts and suggested a much less intensive activity of the TAA1-YUC IAA biosynthesis pathway in this plant than the one in Arabidopsis [73]. However, understanding the auxin biosynthesis pathways in *O. sativa* is still not well developed and further analyses are needed to reach a conclusion about the SE-associated auxin production in the model plants *Arabidopsis* vs. *O. sativa.*

## 5. The Core Regulatory Components of the Auxin-Signalling Pathway That Are Involved in SE

A key insight from the analyses of the SE-related transcriptomes in numerous plants, including wheat [74], palm [32], cotton [70,71,75], Arabidopsis [64,76], camphor tree [77], mangosteen [78], cork oak [79], Indian bean tree [80], coffee [81], maize [11] and lily [82], is that besides auxin biosynthesis other aspects of auxin action including auxin perception, polar transport and response/signalling are also involved in the mechanism of SE induction.

The core components of the auxin-signalling pathway that translate auxin sensing into transcriptional responses include the SKP/CULLIN/F-BOX-ubiquitin (SCF^TIR1^) complex, the Aux/IAA transcriptional co-regulators and the sequence-specific binding proteins, which are called ARFs. The SCF^TIR1^complex comprises TIR1, the AUXIN-SIGNALLING F-BOX1 (AFB1), AFB2 and AFB3 proteins, which are members of the clade of the auxin receptors family (TAARs). TAARs are a component of the SCF-ligase complexes, which direct the members of the AUX/IAA transcriptional repressor protein family to a proteasome-dependent degradation. The degradation of AUX/IAA releases the TOPLESS (TPL) transcriptional co-repressor and allows the ARF proteins to bind at the promoters of the primary auxin responsive genes in order to regulate their transcription [83]. Importantly for the role of auxin in establishing the SE-transcriptome in the explant cells, the specific expression pattern of the auxin-responsive genes is likely due to the specific cellular concentrations of several of the Aux/IAA proteins that seem to result from the auxin content in a cell. However, a quantitative relationship between cellular auxin concentration and gene activity has not yet been indicated, and whether there are separable gene sets that only respond to low or high auxin concentrations remains to be determined [69].

Auxin regulates the specific interactions between the Aux/IAAs and ARFs and almost all of the Aux/IAA proteins can interact with a subset of the ARFs with a long, Q-rich middle region (activating ARFs), whereas interactions with other ARFs are both limited and more specific [84]. In Arabidopsis, 29 and 23 members of the *AUX/IAA* and *ARF* gene families have been identified, respectively, and specific AUX/IAA-ARF interactions are assumed to control the auxin-dependent developmental processes including ZE [84,85]. Therefore, identifying the spatiotemporal and cell-specific sets of the AUX/IAA elements and the interacting ARFs that contribute to the embryogenic transition of somatic cells is required for deciphering auxin-mediated SE induction.

The transcripts encoding AUX/IAAs and ARFs are highly represented in the SE-transcriptomes of Arabidopsis [76,86] and other plants, including cyclamen [87], wheat [74], palm [32,88], cotton [71], camphor tree [77], mangosteen [78], cork oak [79], Indian bean tree [80], coffee [81], maize [11] and lily [82]. Insight into the SE-transcriptomes revealed that the expression profiles of the *AUX/IAA* and *ARF* genes are highly diverse and that the same gene might show both a significant up- and down-regulation depending on the embryogenic system (Appendix A). The high degree of the diversity of the expression patterns of the *ARF* genes reflect the complexity of the transcriptional regulation of these genes, which are fine-tuned by numerous factors including the hormonal status of a tissue [84]. The direct regulation of the SE-expressed *ARF*s by auxin might be considered and an auxin-responsive AuxRE module was found in the majority (84%) of the SE-modulated *ARF*s in Arabidopsis, including *ARF5, ARF6, ARF8, ARF10* and *ARF16* [37].

In contrast to the numerous SE-transcriptomic data, functional analyses of the specific auxin-signalling genes that are engaged in SE are still rare and limited almost exclusively to Arabidopsis. A major drawback for such an analysis seems to be the high functional redundancy within members of the TAAR, AUX/IAA and ARF families and thus, the frequently unobvious and pleiotropic phenotype of the relevant mutant plants [84,85]. Analysis of plants with impaired TAAR receptors of auxin, the *tir1*-*1* and *afb2*-*3* mutants, indicated their hyposensitivity to 2,4-D treatment and a reduced embryogenic response in vitro and the *TIR1* and *AFB2* genes were also postulated to control SE induction via an miR393-mediated mechanism [89]. In addition to Arabidopsis, the down-regulation of *TIR1* and *AFB2* was also observed during SE in cotton but the regulatory pathway that controls their expression has not yet been identified [70,75,90]. Considering that the TAAR proteins have a different binding affinity to auxin herbicides [91], the high binding affinity of the SE-involved TIR1 and AFB2 to the 2,4-D that is used for SE induction in Arabidopsis might be expected [89].

The mutations in the *IAA16, IAA29, IAA30* and *IAA31* genes that resulted in an impaired embryogenic response of the Arabidopsis explants suggests the engagement of these genes in SE induction [76,86]. The expression of the SE-involved *AUX/IAA* genes seems to be regulated by the TFs that have a master function in the embryogenic transition, and relevantly, AGAMOUS-LIKE15 (AGL15) controls the expression of *IAA16* and *IAA30*, while LEAFY COTYLEDON2 (LEC2), which interacts with AGL15 in a feedback regulatory loop (see next paragraph), directly targets *IAA30* [86,92].

In accordance with SE-transcriptome data that suggest that the majority of the *ARF* genes seem to be engaged in controlling SE (Appendix A), numerous *arf* mutants *arf1-3, arf3, arf5-8, arf6, arf7, arf8* were found to be impaired in their embryogenic response [37]. It is noteworthy that the mutants in the *ARF* genes of the contrasting SE-expression profiles were found to display a reduced embryogenic response, and accordingly, the mutations in *ARF5/ARF6* and *ARF3* with an up- and down-regulated expression in SE, respectively, were indicated to negatively affect SE induction [37]. The lack of a direct relationship between the *ARF* gene expression profile and the phenotype of the relevant *arf* mutant implies the complexity of the factors that fine tune the ARF-mediated responses during SE induction. These factors include the distinct spatio-temporal and hormone-dependent regulation of *ARF* gene expression and the complex regulatory interactions within the ARF family members and between the ARFs and other components of the auxin-signalling pathway [37,84].

Several pieces of evidence suggest that similar to ZE (reviewed in [93]), *ARF5*, which encodes the MONOPTEROS (MP) protein, might have a fundamental role in regulating different auxin-controlled aspects of SE. Accordingly, *ARF5* was observed to be the most highly up-regulated member of the *ARF* family during SE induction and the *arf5* mutant had a significantly reduced capacity for an embryogenic response [37]. One of the functions of ARF5 in SE seems to be connected with regulating *PHABULOSA* (*PHB*), which is a positive regulator of *LEC2* in ZE [94] and in support of this assumption, in contrast to the up-regulated expression of *PHB* and *LEC2* in a WT embryogenic culture, an *arf5* mutant culture displayed a significantly reduced *LEC2* transcript level [37,95]. A regulatory ARF5-PHB connection was observed in vivo [96]. Thus, the ARF5 protein was postulated as controlling the TAA1-YUC-mediated pathway of auxin biosynthesis that operates in SE induction by regulating *PHB* and *LEC2* [36]. In line with this hypothesis, ARF5 targets the components of this pathway (*TAA1* and *YUC1, YUC8*) in order to transcriptionally initiate the embryonic ground tissue in an early zygotic embryo [97]. ARF5 might also contribute to the in vitro-induced embryogenic development by controlling the other *TF* genes that have a documented activity in SE, including *HOMEOBOX GENE8* (*ATHB8*) and *TARGET OF MONOPTEROS3* (*TMO3*), *TMO5, TMO6* and *TMO7* [64,76]. In support of this assumption, the regulatory relationships between these TFs and ARF5 were identified in planta [98,99]. Interestingly, the *TMO7*, which is a direct ARF5 target encoding a small TF, moves intercellularly to control the embryonic root specification [99]. Hence, it is tempting to question whether ARF5 also controls the mobility of TOM7 during SE induction and how the ARF5-TOM7 regulation could contribute to establishing the embryonic identity in the somatic cells of an explant.

Many members of the Aux/IAA family have been shown to interact with ARF5 in the development of Arabidopsis in planta [100] and among them is IAA30 of indicated involvement in SE [76,92]. The possible function of ARF5 may also involve regulating the *PIN* genes, which control the polar auxin transport during in vivo development [101,102] and in support of this statement, a *pin1-7* mutant had a reduced embryogenic induction in vitro [103] and the 35S::ARF5 line, which phenocopies the *pin1* mutant [104], was found to be incapable of inducing SE [37]. In order to stimulate the expression of *PIN1* and to control early zygotic embryo patterning, ARF5 cooperates with the WUSCHEL-RELATED HOMEOBOX (WOX) TF gene family members, including WOX1, WOX2, WOX5, WOX8 and WOX9 [105,106,107]. Although the expression of the *WOX2*, *WOX8* and *WOX9* genes was reported in an embryogenic culture of several plants, including grapevine, spruce, Arabidopsis, cotton, barrelclover, larch and longan [64,75,108,109,110,111,112,113,114], the auxin-signalling regulators that control the *WOX* genes in SE induction have not been studied as yet. In particular, *WOX2* seems to be an interesting ARF5 target candidate in regulating SE because of its role in balancing the cytokinin versus the auxin hormone pathways during the establishment of the shoot identity in the zygotic embryo [115]. Relevantly, *ARF10*, which has a positive regulatory impact on de novo shoot regeneration via the activation of the shoot meristem-specific genes, was highly expressed in the SE of Arabidopsis [37,116]. Therefore, ARF10 together with ARF16 and miR160 have been postulated to contribute to the LEC2-controlled auxin biosynthesis in SE-induced tissue [95]. Besides LEC2, other SE-related candidate targets of ARF10/ARF16 involve the *PLETHORA* (*PLT*) genes, which have an essential function in integrating the hormonal inputs during many developmental processes in plants including the auxin responses in the early embryogenic induction (reviewed in [117]). Because *ARF10* and *ARF16* act upstream of the *PLT*s to regulate the stem cell differentiation in Arabidopsis roots [118], the significant accumulation of *ARF10* and *ARF16* transcripts was accompanied by an increased expression of *PLT1* and *PLT2* in an embryogenic culture of Arabidopsis [37,64,76].

Data on *ARF* expression profiling in the SE of Arabidopsis and other plants also suggests that ARF6, ARF7, ARF8, ARF9 and ARF19 (Appendix A) play a role in controlling SE induction; however, their specific functions and the SE-involved targets have not yet been identified.

## 6. Complex Interactions Between the *TF* Genes that Control Auxin-Induced SE

The transcripts of the *TF* genes that are related to hormone responses, in particular those that control auxin metabolism and signalling are overrepresented in the SE-transcriptomes of many plants (reviewed in [119]). In Arabidopsis, over 40% of the SE-modulated *TF* genes have been annotated to hormone-related pathways, primarily to the auxin metabolism and signalling pathways [76]. The best recognised TF-regulatory network that controls SE induction includes the *LAFL* group of genes (for *LEC1/L1L*, *ABSCISIC ACID INSENSITIVE3* (*ABI3*), *FUSCA3* (*FUS3*) and *LEC2*), which have a major and redundant regulatory function in controlling the identity and maturation of a zygotic embryo [120,121]. The overexpression of two of the *LAFL* genes, *LEC1* and *LEC2,* in Arabidopsis seedlings resulted in somatic embryo formation, while the *lec1* and *lec2* mutations significantly inhibited the embryogenic response of the in vitro cultured explants [122,123,124]. *LEC1* and *LEC2* gene expression is co-localised with the SE-involved parts of the 2,4-D-treated explants [125,126,127]. Importantly, *LEC1* and *LEC2* seem to have a common SE-promoting function in plants and the overexpression of these *TFs* has been suggested as an efficient method for improving the embryogenic response in several crop species, including cassava, rapeseed, tobacco and cacao [128,129,130,131]. Taken together, these observations led to a conclusion on universal and essential function of LEC1/LEC2 in controlling both the somatic and zygotic embryonic development in higher plants (reviewed in [132]).

The diversity of the LEC1 and LEC2 targets implies that these proteins contribute to the regulation of various auxin-related processes, which involve the control of auxin biosynthesis (see the paragraph ‘Biosynthesis and Accumulation of Auxin During SE’) and relevantly, members of the Aux/IAA family were postulated to be under the control of LEC1 (*IAA5, IAA16* and *IAA19*) and LEC2 (*IAA1, IAA17, IAA30* and *IAA31*) during embryonic development [92,133,134]. The impact of LEC2 on auxin polar transport also cannot be ruled out because the upregulation of the auxin efflux facilitators, *PIN1* and *PIN2*, was observed in the transgenic tobacco plants that over-expressed *LEC2* [129]. Moreover, the auxin-responsive AuxRE motif, which is present in the *LEC1* and *LEC2* promotors, is evidence of the involvement of the ARF proteins in controlling the auxin-stimulated expression of these genes [37]. The candidate ARFs that control LEC1/LEC2 in the SE-induced explants might be searched for among the *ARFs* that have an up-regulated expression in embryogenic cultures (Appendix A).

In addition to their auxin-related function, the *LEC* genes (*LEC1* and *LEC2* together with *FUS3*) seem to control the ability of explants to respond to SE via the control of the gibberellins (GA)/ abscisic acid (ABA) balance in tissue. Accordingly, *AGL15*, which is an activator of the *GA2ox6* gene that encodes an enzyme that inactivates bioactive gibberellin, was identified among the direct targets of LEC2 [92,132,135]. *AGL15* encodes a MADS box TF that increases the embryogenic competency of tissue when it is ectopically expressed [136]. The expression of *AGL15* is autoregulated [137] and regulated by feedback interactions with other TFs, which involve the direct regulation of *AGL15* by LEC2 and FUS3 and the control of *LEC2* and *FUS3* by AGL15 [86,92,138,139].

Insights into the AGL15-overexpression transcriptomes of Arabidopsis and soybean showed that AGL15 might enhance the embryogenic competency of the explants by promoting tissue dedifferentiation [140] and also revealed the complex interactions of AGL15 with the hormone pathways including auxin signalling [141]. Accordingly, AGL15 was demonstrated to negatively regulate the auxin-signalling components, including the direct repression of *ARF6* and *TIR1* ubiquitin ligase, the indirect repression of *ARF8* and the direct up-regulation of *IAA30* [86,141].

The SE-regulatory network of TFs also involves members of the AINTEGUMENTA-LIKE (AIL) family of the APETALA2/ETHYLENE RESPONSE FACTOR (AP2/ERF) domain TFs [117]. The small AIL family comprises eight members, including the AINTEGUMENTA (ANT), AIL, BABY BOOM (BBM/PLT4) and five other PLT proteins that have overlapping functions in the dividing cells of meristematic and embryonic tissue [142,143]. Of all of the AIL TFs, a central position in regulating SE has been attributed to BBM/PLT4; however, other PLTs (PLT1, PLT2, PLT3, EMK/PLT5 and PLT7) also promote embryogenic induction in response to its overexpression [121,144,145]. Recently, PLT1/PLT2 was reported to control the *LEC2* function in auxin-induced SE and the miR396–*GRF* module has been postulated to control the PLT2-LEC2 interaction [146].

To induce SE, *LEC1* and *LEC2* are positively regulated by BBM, which is another major TF protein that controls the identity and totipotency of the plant embryo, and the SE-promoting activity of BBM requires the LAFL/AGL15 proteins [147]. Knowledge about the other functionally characterised targets of BBM is limited. In search of direct BBM targets, a set of mostly uncharacterised genes that have assumed functions in transcription, cellular signalling and cell wall biosynthesis was identified including the *ACTIN DEPOLYMERISING FACTOR9* (*ADF9*) gene, which encodes an ADF/cofilin protein [148]. Auxin-related genes were revealed within the BBM targets that are involved in SE in Arabidopsis including the genes that are associated with the biosynthesis (*TAA1, YUC3* and *YUC8*), transport (*PIN1* and *PIN4*) and signalling of auxin (*ARF2, ARF6, ARF10, IAA2, IAA7* and *IAA28*) [121]. In addition, a possible role of PLTs in the auxin-related mechanism of SE induction implies the activation of the *YUC* genes (*YUC1* and *YUC4*) by PLT3, EMK/PLT5 and PLT7 in order to stimulate auxin biosynthesis in phyllotaxis [149].

The activators of the LEC-mediated SE pathway also involve the class III HOMEODOMAIN LEUCINE ZIPPER (HD-ZIP III) TFs, PHB and PHAVOLUTA (PHV), which activate *LEC2* in the vegetative development of Arabidopsis [94]. Similarly, PHB/PHV and LEC2 might interact in SE through a positive feedback loop, which, under the repression of miR166, was postulated to control the YUC-pathway of auxin biosynthesis in explants that were undergoing embryogenic induction [95].

Several TFs have been recognised to negatively control the *LAFL* genes. The PICKLE (PKL) protein of the CHD3-type ATP-dependent SWI/SNF chromatin-remodelling factors impacts the global H3K27me3 levels including *LEC1* and *LEC2* [150,151]. In addition, the VIVIPAROUS1/ABI3-LIKE (VAL)/HIGH-LEVEL (VAL1) and VAL2 proteins that like ABI3, FUS3 and LEC2 have a B3 domain were postulated to function as global regulators of the LEC1/B3 gene system, which controls embryo identity in seedlings [152]. The interactions of the VALs with the chromatin factors and the gene repression mechanism that involves the VAL-mediated recruitment of HDAC histone deacetylases and Polycomb group proteins to the target loci were recognised [153,154].

An SE-promoting capacity was also indicated for *WUSCHEL* (*WUS*), which is a member of the *WOX* superfamily of the genes encoding the plant-specific homeobox TFs that are involved in promoting cell division and/or preventing the premature cell differentiation of the stem cells in SAM (reviewed in [155,156]). The positive impact of *WUS* on the embryogenic potential of tissue was indicated given that the gain-of-function mutation in *WUS* led to the spontaneous formation of somatic embryos on Arabidopsis root explants [157] and an overexpression of *WUS* promoted or enhanced the embryogenic response in other plants [158,159,160]. Auxin treatment was found to be essential for correctly regulating the *WUS* expression and the *WUS* transcripts that were detected in a group of embryogenic callus cells were postulated to indicate an auxin accumulation in the SAM and the pro-embryo [103]. Although the ectopic expression of *AtWUS* in transgenic cotton callus was associated with the upregulation of the auxin biosynthesis-regulators, the *LEC1* and *LEC2* genes [160], the content of IAA was not modulated [161]. Hence, the contribution of WUS to the TF-regulatory network of SE seems to be unrelated to auxin biosynthesis and in support of this assumption, within the numerous auxin-related targets of *WUS*, the genes that are involved in auxin biosynthesis have not been identified [162]. In the mechanism of the WUS-mediated control of stem cells in SAM, WUS acts as an auxin rheostat that controls the target loci including numerous genes of the auxin-signalling pathway by regulating histone acetylation [162].

Insights into ZE regulation suggest that in addition to *WUS*, other members of the *WOX* genes family might also interact with the LEC-controlled pathway of SE induction. Accordingly, during stem cell initiation in early zygotic embryo, the *WOX* genes (*WOX2* and its paralogs *WOX1, WOX3* and *WOX5*) were shown to positively control the PHB/PHV TFs that activate the *LEC* genes in both zygotic and somatic embryogenesis [95,115]. Similar to ZE, the regulatory interaction between the WOXs and PHB/PHV might be expected during somatic embryo formation because the expression of both *WOX*5 and *PHB/PHV* was indicated in the embryogenic and callus cells of Arabidopsis [163]. Moreover, the *WOX2*, *WOX8* and *WOX9* transcripts accompany different stages of somatic embryo development in different plants [64,75,108,109,110,111,112,113,114]. However, the regulatory interactions between the *WOX* genes and the LEC-controlled SE pathway require experimental validation.

## 7. Conclusions and Future Prospects

In the past ten years, omics data analysis together with the candidate gene approach have significantly contributed to identifying the regulatory interactions between the TFs that have a key function in SE and the phytohormones, mainly auxin. A revealing picture of the SE-regulatory gene network reveals a high level of the complexity of the extensive crosstalk between the SE-TFs and auxin during the auxin- inducted embryogenic development in somatic plant tissue (Figure 1).

Significant progress in establishing the genetic network that controls SE has been observed and both the down-stream targets and up-stream regulators of the SE-involved TFs have been identified and their functions in relation to auxin have been revealed. However, knowledge about the role of auxin in the regulatory network that controls SE induction is still fragmentary since in addition to gene transcription, auxin regulates a plethora of other cellular processes, including the epigenetic state of chromatin, protein stability/protein-protein interactions and metabolic pathways. Moreover, the interactions of auxin with other hormone pathways, in particular cytokinins, need to be explored in cells that are undergoing SE induction. Since only a small subset of explant cells is capable of embryogenic transition, the methods that enable a single-cell resolution analysis would provide new insights into the auxin-mediated regulatory network that governs the unique developmental plasticity that is demonstrated by somatic plant cells.

The ongoing debate on the SE regulatory network also concerns the degree of the molecular similarity of the embryogenic development that is induced in vitro in somatic tissue to the ZE process. In accordance with the common belief of a close similarity of the regulatory pathways that control both types of plant embryogenesis, several master regulators of ZE, including LEC, PLT and WOX TFs, were found to play a decisive role in the embryogenic reprogramming of plant somatic cells. Moreover, the auxin-related processes including auxin biosynthesis and signalling was placed in centre of SE and ZE regulation. However, recent RNAseq data of an embryogenic culture of Arabidopsis and *Pinus pinaster* showed that the SE-transcriptome seems to be distinctly different from the transcriptome of a zygotic embryo [170,171]. This unexpected finding suggests that the molecular events that trigger SE induction might differ from the ones that operate during ZE and points to the epigenetic processes that orchestrate the transcriptomes of in vitro cultured somatic cells in response to plethora of endo- and exogenous factors, thus making them different from those found in ZE. Therefore, identifying the epigenetic processes that contribute to the embryogenic transition in somatic plant cells in response to auxin would lead to further progress in revealing the SE regulatory network.

## Figures and Tables

**Figure 1 ijms-21-01333-f001:**
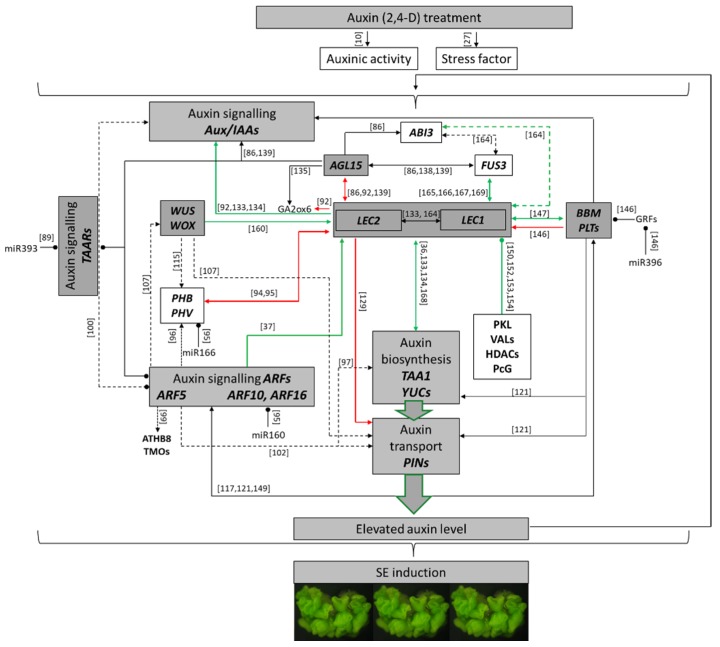
An overview of the regulatory interactions that control SE induction. Treating explants with auxin (mostly 2,4-D) activates a number of the *TF* genes that have a regulatory function in embryonic development, including the *LEC1* and *LEC2* of the *LAFL* group, *BBM* (*PLTs*), *AGL15* and *WUS/WOX* genes. The down-stream targets include the genes of the TAA1/YUC of the auxin biosynthesis pathway, the PINs that control auxin transport and the core regulators of auxin signalling (AUX/IAAs and ARFs). Other regulatory elements of the SE-network also involve miRNAs and the epigenetic regulators (PKL, VALs, HDACs). The arrows and circle-shaped ends indicate the activation or repression of gene expression, respectively. The solid and dashed lines indicate the experimentally validated and suggested regulatory interactions, respectively. The red and green lines indicate the regulatory interaction between LEC2 and LECs, respectively [164,165,166,167,168,169].

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
