# Peer review of "Current Perspectives on the Auxin-Mediated Genetic Network that Controls the Induction of Somatic Embryogenesis in Plants"

_ijms, 2020, doi:10.3390/ijms21041333_

Round 1
Reviewer 1 Report
In « Current perspectives on the auxin-mediated genetic network that controls the induction of somatic embryogenesis in plants” the Authors report on an important subject in plant physiology.
The molecular mechanisms here described are of interest for the readership of IJMS, but also for the specialists. The need for a review on this subject is unquestionable.
I'm very impressed with the conciseness and the precision with which the authors present these complex concepts and signaling pathways.
I unreservedly recommend this work for publication in its present form.
Author Response
Thank you so much for this great and positive review.
Reviewer 2 Report
Wójcikowska et al. review somatic embryogenesis (SE) and auxin’s role for induction of SE. They primarily focus on transcriptional studies of a subset of transcriptional regulators including auxin receptor signaling components and also mention auxin biosynthesis pathways. They cover the involvement of the other growth regulators such as cytokinin and GA. Development and use of SE have been practiced for many years in clonal propagation of plants. Although the techniques have been applied in biotechnology, the mechanisms for inducing SE are not well understood to date. Auxin is one of the essential components which drive SE induction. The authors have piled the past and current literature and gathered a collection of transcription factors(TF) IDs reported to respond during SE.
In table 1, a list of ARF and IAA genes and their expression profiles during SE are shown. Some are induced and some are decreased. But regardless of the directions of the changes in transcriptional profiling, the genetic mutants of the corresponding genes appear always showed reduced SE induction. For example, ARF5 (MP) expression increase, and thus one expects this gene has a role in SE induction. Consistently, its mutant has reduced SE inducing capability. However, when we look at ARF3, whose expression decreases during SE, one expects to see its mutant possibly having increased capability in SE induction. But the observation is the opposite and its mutation also caused decreased SE induction. Transcriptional changes are often transient and thus the reported results can vary depending on what time frame these samples were analyzed. Could the authors add some insights to that?
Line 65, 2,4-D physiological effects as a herbicide and an SE inducer. They brought up some intriguing facts. Monocot is relatively insensitive to 2,4-D herbicide and thus 2,4-D can be used for broadleaf plant control. It might be useful to comment to the sensitivity of monocot to 2,4-D in SE induction. They included wheat and maize as plant species in which auxin related gene transcriptional reprogramming occur during SE (line 185). Should one expect the target of 2,4-D in SE induction being differ from that of the overdose effect? It might be useful mention the concentrations of 2,4-D commonly used for SE induction experiments and those used as herbicide to give an idea to what the “overdose” indicates.
They mention some similarity of transcriptional responses, auxin accumulation or the responding gene IDs between zygotic embryogenesis (ZE) and SE. ARF5 (around line 214), WUS (line 360), and YUCCA auxin biosynthesis gene (around line 128) sound key to both SE and ZE. Could they mention if there are quantitative measurements, such as the presence of the whole genome wide comparison like transcriptome proofing, for commenting the similarity between SE and ZE?
Minor points:
Author affiliation. Please include the county where University of Silesia is located. In general there are lots of abbreviated words and it can be hard to follow. Having the list for abbreviation in the end of this manuscript as they did is really helpful. But I still could not find what “IZE” means in line 136 and 138. Also please fully spell “TSA” in line 138 since it appears only once. 1. they made a nice summary chart for auxin signaling in SE development. Each genes are connected as “regulatory interaction.” To make it consistent and clear, could they add “regulatory” before “interaction” in the following sentences?: Black and dashed lines indicate the experimentally validated and suggested (insert here) interactions, respectively. Red and green lines indicate the (insert here) interaction between LEC2 and LECs, respectively. When we talk about interaction, it can mean protein-protein interaction, genetic interaction, and physiological interaction. In this case, I believe interaction at gene expression level.
Author Response
Responses to the comments of the Reviewer 2
In table 1, a list of ARF and IAA genes and their expression profiles during SE are shown. Some are induced and some are decreased. But regardless of the directions of the changes in transcriptional profiling, the genetic mutants of the corresponding genes appear always showed reduced SE induction. For example, ARF5 (MP) expression increase, and thus one expects this gene has a role in SE induction. Consistently, its mutant has reduced SE inducing capability. However, when we look at ARF3, whose expression decreases during SE, one expects to see its mutant possibly having increased capability in SE induction. But the observation is the opposite and its mutation also caused decreased SE induction. Transcriptional changes are often transient and thus the reported results can vary depending on what time frame these samples were analyzed. Could the authors add some insights to that?
Response:
Following the reviewer comment we have added a short paragraph that refer to SE-phenotypes of the arf mutants and their relation to the ARF expression profiles in SE that are shown in Table S2 (lines: 228-238).
Line 65, 2,4-D physiological effects as a herbicide and an SE inducer. They brought up some intriguing facts. Monocot is relatively insensitive to 2,4-D herbicide and thus 2,4-D can be used for broadleaf plant control. It might be useful to comment to the sensitivity of monocot to 2,4-D in SE induction. They included wheat and maize as plant species in which auxin related gene transcriptional reprogramming occur during SE (line 185). Should one expect the target of 2,4-D in SE induction being differ from that of the overdose effect? It might be useful mention the concentrations of 2,4-D commonly used for SE induction experiments and those used as herbicide to give an idea to what the “overdose” indicates.
Response:
We have followed the reviewer comment on diverse and concentration-dependent effects of 2,4-D in plants that grow in the field vs in in vitro cultures. Information on concentrations of 2,4-D effective in the field vs applied for SE-induction in mono- and dicots have been provided (lines 69-77).
We think, the targets of 2,4-D in SE induction to be quite different to that induced by the high 2,4-D concentrations that are lethal. In our opinion, the transcriptomes induced by the high 2,4-D concentrations would not reflect a regulatory function of 2,4-D in auxin signaling pathway (as it is expected for the low 2,4-D concentrations used in vitro) but rather results from the global deregulation of the cell transcriptome responding to the lethal effects/processes.
They mention some similarity of transcriptional responses, auxin accumulation or the responding gene IDs between zygotic embryogenesis (ZE) and SE. ARF5 (around line 214), WUS (line 360), and YUCCA auxin biosynthesis gene (around line 128) sound key to both SE and ZE. Could they mention if there are quantitative measurements, such as the presence of the whole genome wide comparison like transcriptome proofing, for commenting the similarity between SE and ZE?
Response
Discussion on the similarity between SE and ZE continues since the early nineties of XX century and numerous reports have provided evidences supporting the common believe that SE resembles its zygotic counterpart in some aspects including the regulatory TF genes that trigger the embryogenic development. However, a recent work on SE vs ZE transcriptomes unexpectedly indicated the significant differences between them (Hofmann et al. 2019) and suggests that the molecular events triggering the SE induction might differ from that operated during ZE. Relevantly, the epigenetic processes that in response to plethora of endo- and exogenous factors orchestrate the somatic cell transcriptomes seem to be different to that operating during ZE. Thus, identification of the epigenetic processes that contribute to the embryogenic transition in somatic plant cells in response to auxin determines further progress in revealing of the SE regulatory network.
Accordingly, a short paragraph has been added to the Conclusions (lines 423-437).
Author affiliation. Please include the county where University of Silesia is located.
Response:
We have added Poland, the country where University of Silesia is located.
In general there are lots of abbreviated words and it can be hard to follow. Having the list for abbreviation in the end of this manuscript as they did is really helpful. But I still could not find what “IZE” means in line 136 and 138.
Response:
We are sorry for that, we have changed the manuscript in line (147) and the abbreviation list has been corrected (line 455).
Also please fully spell “TSA” in line 138 since it appears only once.
Response:
It has been corrected (line 148)
They made a nice summary chart for auxin signaling in SE development. Each genes are connected as “regulatory interaction.” To make it consistent and clear, could they add “regulatory” before “interaction” in the following sentences?: Black and dashed lines indicate the experimentally validated and suggested (insert here) interactions, respectively. Red and green lines indicate the (insert here) interaction between LEC2 and LECs, respectively. When we talk about interaction, it can mean protein-protein interaction, genetic interaction, and physiological interaction. In this case, I believe interaction at gene expression level.
Response:
We have changed the Figure 1 description as followed:
Arrows and circle-shaped ends indicate the activation or repression of gene expression, respectively. Solid and dashed lines indicate the experimentally validated and suggested regulatory interactions, respectively. Red and green lines indicate the regulatory interaction between LEC2 and LECs, respectively (lines: 410, 411). In this case, we talk about interaction at gene expression level.